# Short Term Presence of Subretinal Fluid in Central Serous Chorioretinopathy Affects Retinal Thickness and Function

**DOI:** 10.3390/jcm9113429

**Published:** 2020-10-26

**Authors:** Maciej Gawęcki, Agnieszka Jaszczuk, Andrzej Grzybowski

**Affiliations:** 1Dobry Wzrok Ophthalmological Clinic, Kliniczna 1B/2, 80-402 Gdansk, Poland; jaszczuk.amt@gmail.com; 2Department of Ophthalmology, University of Warmia and Mazury, 10-719 Olsztyn, Poland; ae.grzybowski@gmail.com; 3Institute for Research in Ophthalmology, 60-554 Poznan, Poland

**Keywords:** central serous chorioretinopathy, subthreshold diode micropulse, photodynamic therapy, spectral-domain optical coherence tomography

## Abstract

*Background:* Acute central serous chorioretinopathy (CSCR), with subretinal fluid (SRF) resolving spontaneously within a few months from disease onset, has been considered as a benign and self-limiting disease for many years. This study sought to discover if a short presence of SRF can result in morphological and functional damage to the retina. *Materials and methods:* The study included patients treated by subthreshold diode micropulse laser (SDM) application for acute CSCR at the Dobry Wzrok Ophthalmological Clinic between January 2018 and November 2019. Inclusion criteria were: first episode of CSCR; duration of symptoms of two months or less; complete resolution of subretinal fluid (SRF) after a single session of SDM; and a lack of any retinal pathology, previous CSCR episode, significant anisometropia or amblyopia in the collateral eye. Fifteen patients fulfilled the inclusion criteria, including 13 males and two females aged 42.3 ± 9.5 years. The mean duration of symptoms before treatment was 4.7 ± 1.3 weeks on average. Baseline and follow-up examinations were performed in both the affected and collateral eyes and included best-corrected visual acuity (BCVA); spectral-domain optical coherent tomography measurements such as central retinal thickness (CRT) and minimal foveal thickness (MFT) (at the follow-up visit only); fluorescein angiography (at presentation only) and fundus autofluorescence. The first follow-up visit, when the total resolution of SRF was noted, was conducted between 8 and 12 weeks after SDM. *Results:* Resolved CSCR eyes had significantly poorer BCVA, CRT, and MFT findings in comparison with healthy collateral eyes (respectively, 0.11 +/− 0.1 vs. 0.01 +/− 0.04 logMAR; 238.80 +/− 23.39 vs. 264.87 +/− 21.22 µm and 178.93 +/− 16.88 vs. 199.47 +/− 17.87 µm) despite the short period of CSCR duration (maximum of 14 ± 2.15 weeks on average). *Conclusion:* Short presence of SRF typical for acute CSCR can affect retinal function and morphology resulting in poorer visual outcome.

## 1. Introduction

This retrospective study sought to determine whether the short term presence of subretinal fluid (SRF), typical for resolving acute form of central serous chorioretinopathy (CSCR), results in functional or morphological damage to the retina. SRF is a major symptom of acute and chronic forms of CSCR. Acute CSCR is usually defined as lasting up to 4 months, while the chronic form as lasting longer than that. For many years, acute CSCR has been regarded as a self-limiting and benign disease, with the majority of patients recovering without any damage within the first three to four months after disease onset [1,2,3]. Because of the risks of conventional macular photocoagulation, no treatment was traditionally recommended during this period [4,5]. This belief and therapeutic approach are still widely retained in ophthalmological practice. On the other hand, it is also widely acknowledged that the chronic form of CSCR can result in permanent and significant damage to the retina, expressed as a loss of photoreceptors, retinal thinning, and impaired visual acuity. The mechanism of the damage is attributed to the prolonged presence of SRF and separation of the photoreceptors from the retinal pigment epithelium (RPE). SRF prevents nutrients to be transported from the choroid to the outer retina. Additionally, shedding of photoreceptor outer segments is compromised, which results in photoreceptor damage and accumulation of debris under the sensory retina. Retinal damage in chronic CSCR is confirmed by many studies and not questioned [6,7,8]. What is not determined yet, however, is the precise time point of retinal injury, especially as there is no linear correlation between the duration of CSCR and retinal thinning or visual impairment [9]. Our previous studies on CSCR showed that retinal thinning progresses as the subretinal fluid (SRF) persists [8,9]. Still, we do not know how long can SRF stay unresolved without causing significant impairment of the retina. In the current study, we investigate whether morphological and functional damage can occur early within the first few months during the course of acute and resolved CSCR.

## 2. Materials and Methods

The study was a retrospective one and involved standard procedures and standard consent forms used for every clinical intervention performed in the Dobry Wzrok Ophthalmological Clinic. All procedures performed in this research were conducted in accordance with the ethical standards of the clinic’s institutional research committee and with the principles of the 1964 Declaration of Helsinki.

Records of patients treated for CSCR at Dobry Wzrok Ophthalmological Clinic between January of 2018 and August of 2020 were analyzed. Data from a total of 36 patients presenting with a duration of CSCR symptoms lasting two months or less were extracted. All patients were treated with a subthreshold diode micropulse (SDM) laser at the time of diagnosis, which is a standard first-line treatment for CSCR in the clinic. Of these 36 patients, 15 met the study inclusion criteria, which included first episode of CSCR in one eye, absence of history of CSCR or any other preceding retinal disease in both eyes, absence of amblyopia in either eye, absence of anisometropia larger than 1D in spherical equivalent, and complete resolution of subretinal fluid within 3 months after a single session of SDM. The aim of the study was not to analyze the efficacy of SDM treatment in CSCR, but to evaluate the morphological and functional alterations in the retina that occur after short term presence of subretinal fluid. This concept was based on the assumption of the non-damaging character of SDM treatment, which was confirmed in many previous studies. According to available data, SDM does not affect retinal thickness in any way. [10,11,12,13,14,15] That is why we believe, that cases with resolved SRF after SDM could be treated as a model for CSCR cases that resolve spontaneously. Healthy collateral eyes were treated as a control group with the reference to parameters of retinal thickness and visual acuity.

In total, there were 13 men and two women included in the study group, with an average age of 42.3 ± 9.5 years. The duration of symptoms at the time of presentation ranged from three weeks to eight weeks, with a mean value of 4.7 ± 1.3 weeks. As all patients in the group showed SRF as being resolved at the first follow-up visit after SDM treatment, the maximum possible duration of CSCR was calculated as the duration of symptoms at presentation, plus the period of time to the first follow up visit (mean: 14 ± 2 weeks). Demographics of the study group are summarized in Table 1.

Baseline ophthalmological examinations included best-corrected visual acuity (BCVA) using a logMAR chart, spectral-domain optical coherence tomography (SD-OCT) (Zeiss Cirrus 4000 OCT; Carl Zeiss Meditec AG, Jena, Germany), fluorescein angiography (FA) (Zeiss FF-450; Carl Zeiss Meditec AG, Jena, Germany), and fundus autofluorescence (FAF) (Zeiss FF-450; Carl Zeiss Meditec AG, Jena, Germany). The diagnosis of the active form of CSCR was determined by the presence of SRF on SD-OCT and focal leakage on FA. Preceding episodes of CSCR in collateral eyes were excluded by the fundus examination and FAF photography, which confirmed the absence of retinal pigment epithelium (RPE) abnormalities. The presence of subretinal neovascularisation (CNV) was excluded by FA and OCT angiography. The follow-up visit for each patient was scheduled at a maximum of three months after SDM treatment. The follow-up examination included a BCVA examination using a logMAR chart, SD-OCT imaging, and FAF photography to elucidate any damage to the retina after SDM. An examination was performed on both eyes.

SD-OCT measurements included central retinal thickness (CRT), which refers to the mean retinal thickness within the central circle at the posterior pole of 1 mm in diameter, and minimal foveal thickness (MFT), which refers to the minimal retinal thickness at the foveola. The measurement of CRT and MFT was rational to conduct following the complete resolution of SRF, as, in the active form of the disease, measurements include the portion of the fluid in the foveola and do not reflect the actual thickness of the sensory retina.

SDM was performed with the use of a Supra 577 multispot yellow laser (Quantel Medical, Bozeman, MT, USA) on the basis of SD-OCT mapping. The whole area of retinal edema and beyond, detected on SD-OCT images, was covered with confluent foci of the micropulse laser including the fovea. The SDM parameters were: spot diameter, 160 µm; power, 250 mW; duty cycle, 5%; and pulse duration, 0.2 s.

BCVA and SOCT parameters were compared between the affected resolved eyes and healthy collateral eyes. We treated collateral eyes as a control group. As significant anisometropia and any preceding ocular disorders in collateral eyes were excluded, we believe that they can serve as a reliable reference for detecting possible damage in affected eyes.

### Statistical Analysis

Statistical analysis was performed using the Statistica version 10.0 software program (StatSoft Inc., Tulsa, OK, USA). The following primary parameters of descriptive statistics were calculated: arithmetic mean (M), median (Me), standard deviation (SD), and the minimum (Min) and maximum (Max) values.

Normality of distribution was assessed by Shapiro–Wilk test. Comparisons of parameters before and after treatment as well as of parameters in the affected eyes after treatment versus in collateral healthy eyes were conducted with the use of the Student’s *t*-test or Wilcoxon test depending on the normality of the distribution. The results were quantified as being statistically significant when *p* < 0.05.

## 3. Results

The study group consisted of cases with resolved CSCR, ruled as such when the total resorption of SRF was observable. Besides this anatomical improvement, the retinal morphology and function showed further changes: CRT was significantly reduced with the resolution of SRF and the BCVA was typically improved. Specific parameters before and after treatment are presented in Table 2.

Importantly, improvements in CRT and BCVA do not necessarily indicate a return to normal parameters of morphology and function. This is why the eyes with resolved CSCR were compared with the collateral eyes with no signs of any retinal pathology or amblyopia. As part of this research, CRT, MFT, and BCVA were confronted between these two groups, and Table 3 presents the statistical significance of these differences.

Resolved acute CSCR eyes had significantly thinner central retinas than the collateral nonaffected eyes, including with respect to MFT, which refers to a location with the highest concentration of photoreceptors. BCVA was also poorer, with the difference being close to one line worse on the Snellen chart, in resolved acute CSCR eyes.

Figure 1 and Figure 2 show a comparison of the SD-OCT scans of the affected resolved eyes with collateral eyes. Left eye in Figure 1 and right eye in Figure 2 have reduced CRT when compared to healthy collateral eyes.

## 4. Discussion

Our study shows that even short term presence of SRF in the course of the acute form of CSCR can affect retinal thickness and visual acuity. In the analyzed group, the presence of SRF was probably much shorter than the mean predicted 14 weeks, but still we found statistically significant differences between parameters of the affected resolved eye and collateral eye with no detectable retinal damage (neither in SD-OCT or FAF, nor in the medical history). On average, CRT in previously diseased eyes was 26 µm thinner and BCVA was reduced by 0.1 logMAR in comparison with healthy collateral eyes. As we see retinal thinning is generally more prominent than loss of visual acuity, which has to be confirmed in a larger sample. However, both outcomes are in agreement with prior studies finding visual functional and morphological impairments in CSCR [6,7,8]. The most striking, is the fact that such a tendency is visible so early after the onset of the disease. According to available data, SDM is a safe procedure with no known adverse treatment effects and does not itself affect retinal thickness [10,11,12,13,14,15]. Interaction between retina and micropulse laser in subthreshold mode does not cause cell death or damage. The thermal effect of the laser results in secretion of “heat shock proteins”—cytokines that optimize cell function and prevent apoptosis. [16,17]. So far retinal thinning has not been reported as a complication of SDM treatment in any available study. Thus, according to this knowledge, retinal thinning in the resolved acute CSCR eyes found in our study cannot be attributed to SDM treatment or, at least, such dependence is very unlikely.

We realize, that the duration of the follow up in our study is relatively short and that brings up a question on sustaining morphological and functional results after a longer period of time. In the medical literature we did not find reports on the improvement of retinal thickness after resolved CSCR. Photoreceptor recovery after resolution of SRF is reported, however, very seldom and in very few cases [18]. We believe, that in general, after resolved CSCR we can expect permanent damage rather than spontaneous improvement. Some amount of decline in the quality of vision and contrast sensitivity after CSCR has to be expected practically in every case, despite prompt initiation of treatment and short duration of the disease.

So far, there are very few papers in the medical literature that refer to the potential damaging character of acute CSCR [19]. This is probably due to the prevailing traditional opinion about its benign and unharmful nature. Patients whose CSCR resolves within a short period of time have attracted clinical attention only recently. Nevertheless, several studies have documented both functional and morphological disturbances of the retina following acute CSCR. Hata et al. measured the outer nuclear layer (ONL) in patients with an active form of CSCR lasting more than one month [20]. Thinning of the ONL was noted already in the first month after disease onset and progressed further as the SRF persisted. Collateral eyes showed significantly greater ONL thickness. Baran et al. reported long-term disturbances in the quality of vision after resolved acute CSCR [21]. Despite having 6/6 BCVA, after a few years, patients with prior CSCR demonstrated color-vision defects and some amount of loss in contrast sensitivity. Contrast sensitivity was also impaired after resolved acute CSCR in the study reported by Lourthai et al., although their patients had excellent final BCVA results [22].

The finding of potentially irreversible macular damage early in the course of CSCR, such as the finding that retinal thinning increases with increased duration of SRF, may indicate the value, if not importance, of early intervention to hasten disease resolution. Our findings suggest that waiting for spontaneous remission for even a few months may result in poorer final BCVA and impaired retinal architecture. It is therefore reasonable to take measures to shorten the duration of the disease as much as possible, particularly in light of the safety of SDM and half-fluence photodynamic therapy (PDT).

In the medical literature, a number of papers have analyzed the effects of early treatment in CSCR with the use of different therapeutic modalities. A few studies have advocated for the use of SDM in the acute type of CSCR, finding good anatomical and functional results [23,24,25]. Gawecki et al. revealed better functional results when SDM was applied early on in the course of the disease [23]. Arora et al. compared the outcomes of SDM in acute CSCR with traditional observation, noting better functional results were achieved in patients treated with the former approach [25]. In another study, subthreshold continuous-wave argon laser application for acute CSCR lasting less than one month versus observation found that treated patients improved faster and had better contrast sensitivity at six months of follow-up compared to untreated patients [26]. The results of our present study do not stand contrary to these reports. A loss of approximately 0.1 logMAR in BCVA was noted also in our previous study [23]. Arora et al. report better BCVA of 0.03 logMAR after early SDM treatment in acute CSCR [25], however with such a small study group difference between our study and his, this has to be treated as insignificant.

Despite optimistic results of early treatment of acute CSCR with SDM, it has to be emphasized that most of the published studies are case series, not randomized trials. More research is needed to clearly outline the algorithm of treatment of CSCR with SDM, also with respect to treatment protocols [27].

Laser photocoagulation (LPC) has been used for the treatment of CSCR for many years before the advent of subthreshold treatments. Robertson et al. and Leaver et al. proved that direct LPC shortened the duration of the disease [28,29]. Further research revealed, however, that LPC did not significantly influence final visual outcome nor prevent disease recurrences. The results of the current study suggest this lack of visual improvement following LPC may reflect permanent macular damage already occurring prior to the initiation of treatment, traditionally delayed waiting for spontaneous resolution due to the risks of conventional continuous wave (CW )macular photocoagulation [30]. Other, more modern studies suggest that traditional LPC does not seem to be superior to observation in treating acute CSCR [31].

The other treatment modality used with success in acute CSCR is photodynamic therapy. Studies have reported good functional and morphological effects with improvements in retinal morphology, BCVA, and retinal sensitivity [32,33,34,35,36]. Wu et al. in a randomized placebo-controlled trial showed better improvement in BCVA and multifocal electroretinography in patients treated with half-dose PDT compared to patients receiving a placebo [37]. Chan et al. provided results of 12 months of follow-up of patients with acute CSCR treated with half-dose PDT versus placebo, where both BCVA and the resorption of SRF were significantly better in the treated group [38]. Zhao et al. proved that as low as 30% of a full verteporfin dose in PDT is effective in acute CSCR treatment [39]. On the other hand, Kim et al. revealed faster resorption of SRF in patients with acute CSCR treated by half-dose PDT, although without long-term functional and anatomical benefits [40].

As can be seen from the above analysis, therapies exist that can prove effective in treating acute CSCR. The available research gives evidence that SDM and PDT have potential to shorten the course of the disease and limit retinal damage. However, effects of both treatment modalities have to be placed in the perspective of the visual outcome of self-resolving form of CSCR. That needs further randomized research.

### Limitations

Our study suffers from the weaknesses common to retrospective reports. We realize that our study group is relatively small; however, still, it was large enough to reach statistically significant results. Confirmation of our results in a larger population sample is needed. Besides, we realize, that the study would be more distinct if we could refer results of resolved SDM treated CSCR cases to spontaneously resolved CSCR patients. However, as we strongly believe that early CSCR treatment is beneficial to the patient, collecting cases who were subject to observation only, is in our opinion controversial.

Separately, our study evaluated only BCVA as a marker of retinal function. Further research should include other methods of visual function testing such as microperimetry and mesopic visual function testing, and electroretinography in the study design in order to more precisely characterize functional impairment in acute CSCR and long-term follow to determine to which these early impairments may be permanent or even progressive absent recurrence of submacular fluid.

## 5. Conclusions

Our study shows that the presence of subretinal fluid in the course of CSCR, even in its acute and fast-resolving form, can lead to functional and morphological alterations of the retina. Further research on shortening of the course of the disease is needed to avoid potential damage to the retina.

## Figures and Tables

**Figure 1 jcm-09-03429-f001:**
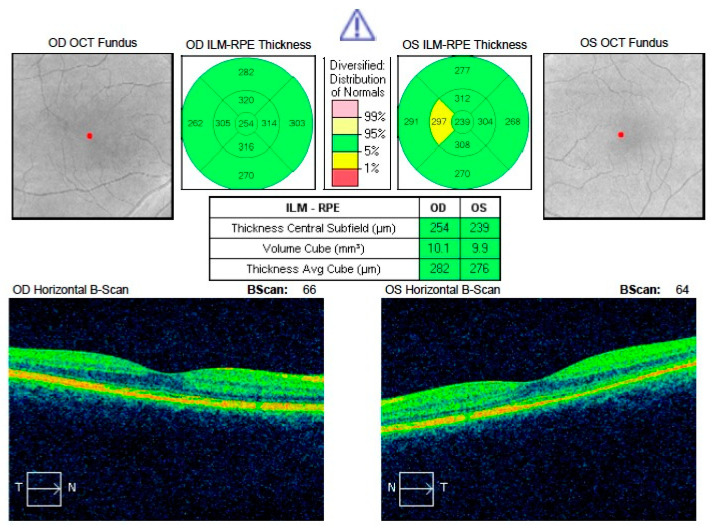
Comparison of CRT in resolved acute central serous chorioretinopathy (CSCR) of the left eye with healthy right eye. Visible reduction in CRT in the left eye. SD OCT: spectral-domain optical coherence tomography; RPE: retinal pigment epithelium; OS—left eye; OD—right eye; OCT—optical coherence tomography; ILM-RPE—internal limiting membrane—retinal pigment epithelium

**Figure 2 jcm-09-03429-f002:**
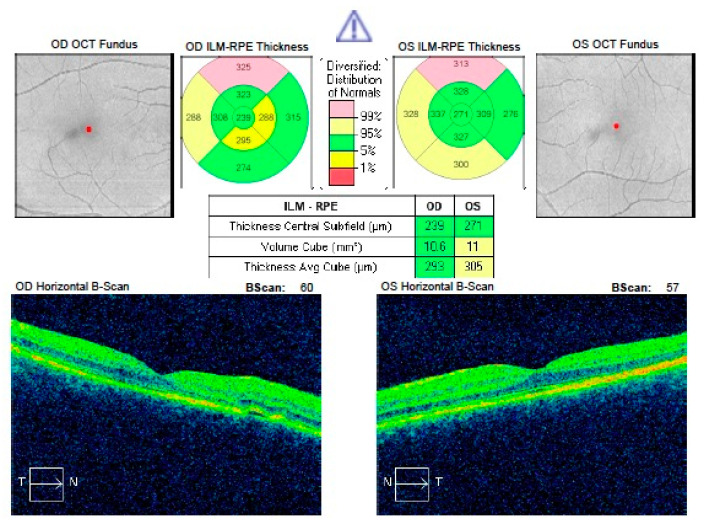
Comparison of CRT in resolved acute CSCR of the right eye with healthy left eye. CRT in the right eye is significantly reduced.

**Table 1 jcm-09-03429-t001:** Demographics of the study group.

Parameter	Mean	Median	Minimum	Maximum	SD
Age	42.07	42.00	30.00	61.00	9.33
Duration of CSCR at presentation (weeks)	4.73	4.00	3.00	8.00	1.33
Maximum time to resolution of SRF (weeks)	14.07	14.00	12.00	18.00	2.15

CSCR: central serous chorioretinopathy, SD: standard deviation, SRF: subretinal fluid.

**Table 2 jcm-09-03429-t002:** Comparison of parameters of the affected eye before and after treatment (*t*-test for CRT and BCVA; Wilcoxon test for MFT).

Parameter	Before Treatment			After Treatment			*p*-Value
	Mean Value	Median Value	SD	Mean Value	Median Value	SD	
CRT (µm)	406.40	403.00	96.62	238.80	240.00	23.39	<0.0001
MFT (µm)	341.08	281.00	123.46	178.93	174.00	16.88	<0.0001
BCVA (logMAR)	0.34	0.30	0.22	0.11	0.1	0.13	<0.0001

CRT: central retinal thickness, MFT: minimal foveal thickness, BCVA: best-corrected visual acuity, SD: standard deviation.

**Table 3 jcm-09-03429-t003:** Comparison between parameters of the affected eye after treatment and the collateral eye (*t*-test for CRT and BCVA; Wilcoxon test for MFT).

Parameter	Affected Eye after Treatment			Collateral Eye			*p*-Value
	Mean Value	Median Value	SD	Mean Value	Median Value	SD	
CRT (µm)	238.80	240.00	23.39	264.87	266.00	21.22	0.000023
MFT (µm)	178.93	174.00	16.88	199.47	194.00	17.87	0.001
BCVA (logMAR)	0.11	0.10	0.13	0.01	0.00	0.04	0.01

CRT: central retinal thickness, MFT: minimal foveal thickness, BCVA: best-corrected visual acuity, SD: standard deviation.

## Data Availability

The datasets generated during and/or analyzed during the current study are available from the corresponding author on reasonable request.

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
