# Peer review of "Short Term Presence of Subretinal Fluid in Central Serous Chorioretinopathy Affects Retinal Thickness and Function"

_jcm, 2020, doi:10.3390/jcm9113429_

Round 1

Reviewer 1 Report

The present study assessed if the short-term presence of subretinal fluid in acute CSC can result in morphological and functional damage to the retina. The authors analyzed patients with acute CSC treated by subthreshold diode micropulse laser (SDM). The main limitation of this study is that it is a retrospective analysis that lacks a control group with active CSC treated with placebo.

Reviewer 2 Report

Although the paper is well written, the main problem are:

  1. The lack of controls in a condition that can spontaneously resolve even in the short term
  2. The short follow-up in a condition that often recovers visual function over a longer period
  3. Although the paper provides the current literature supporting  the possibility of shortening the length of the episode by using subthreshold diode micropulse laser (SDM) there is still no good prospective randomised masked evidence that SDM shortens episodes.
  4. More fundamentally, there is no attempt to isolate the effect of SRF and RPE dysfunction - this could be addressed by comparing the visual recover in cases where there is sub-focal RPE leak compared to those cases where the leak is extra-foveal.
  5. Finally,  it is possible that some of the reduced visual function may be the SDM rather than the SRF and it would take a control group to exclude this

The claim is that even with reduction of length of sub-retinal fluid (or short period of SRF) there is functional and morphological change.

  1. You would have to include a control group that had longer fluid to show there is an advanctage of shortening the episode.
  2. You would have to control with a short-lasting spontaneous groups to show the damage in your cases was not associated with the diode laser.
  3. You would have to make the assessment at 12 months after resolution to get an idea if the changes you demonstrate are long-term or still recovering. This is too short a follow-up

Reviewer 3 Report

Comments to the authors

In this manuscript the authors reported morphological and functional risk for presence of subretinal fluid in acute central serous chorioretinopathy. They retrospectively analyzed visual acuity change and OCT findings after subthreshold diode micropulse laser (SDM) treatment for acute central serous chorioretinopathy (CSCR). They revealed that resolved CSCR eyes had significantly poorer BCVA, CRT, and MFT findings in comparison with healthy collateral eyes. According these results, they concluded short presence of SRF typical for acute CSCR can affect retinal function and morphology.

The authors are better for analyze little more as follows.

Major points

  1. Although they compared BCVA and morphological findings of CSCR eyes with SMD treatment and healthy non-treated collateral eyes, they didn’t refer the effect of micropulse laser to retina. And they concluded that treated CSCR eyes had poorer BCVA and morphological findings because of SRF for CSCR. They should consider the impact of SDM.
  2. They revealed poor functional and morphological findings for CSCR despite of early treatment. In this study, the mean duration of symptoms before treatment was 4.7 weeks. It is very short period in real world. Is it possible to be shorten for observation period in real world ?

Specific Points

  1. In Discussion, authors quoted some studies about the use of SMD in the acute type of CSCR. Some studies reported the good results of BCVA and morphological findings after early SMD treatment. These results are different to authors results. They should describe the reason more.

Round 2

Reviewer 2 Report

Many thanks for addressing the points raised in the original review.  I am happy they have been dealt with.  I am still concerned about the short follow-up but this can be reviewed later by examining the same co-hort after 1 year.